# Biomarkers in Anal Cancer: Current Status in Diagnosis, Disease Progression and Therapeutic Strategies

**DOI:** 10.3390/biomedicines10082029

**Published:** 2022-08-20

**Authors:** Maria Cecília Mathias-Machado, Renata D’Alpino Peixoto, Camila Motta Venchiarutti Moniz, Alexandre A. Jácome

**Affiliations:** 1Department of Gastrointestinal Medical Oncology, Oncoclinicas, São Paulo 04538-132, Brazil; 2Department of Oncology, ICESP—Instituto do Cancer do Estado de São Paulo, University of São Paulo, São Paulo 01246-000, Brazil; 3Department of Gastrointestinal Medical Oncology, Oncoclinicas, Belo Horizonte 34000-000, Brazil

**Keywords:** anal neoplasms, biomarkers, prognostic, predictive, diagnostic, immunotherapy, PD-L1, tumor mutational burden

## Abstract

Squamous cell carcinoma of the anal canal (SCCA) is a rare neoplasm, but with rising incidence rates in the past few decades; it is etiologically linked with the human papillomavirus (HPV) infection and is especially prevalent in immunocompromised patients, mainly those infected with HIV. Fluoropyrimidine-based chemoradiotherapy remains the cornerstone of the treatment of non-metastatic disease, but the locally advanced disease still presents high rates of disease recurrence and systemic therapy of SCCA is an unmet clinical need. Despite sharing common molecular aspects with other HPV-related malignancies, such as cervical and head and neck cancers, SCCA presents specific epigenomic, genomic, and transcriptomic abnormalities, which suggest that genome-guided personalized therapies should be specifically designed for this disease. Actionable mutations are rare in SCCA and immune checkpoint inhibition has not yet been proven useful in an unselected population of patients. Therefore, advances in systemic therapy of SCCA will only be possible with the identification of predictive biomarkers and the subsequent development of targeted therapies or immunotherapeutic approaches that consider the unique tumor microenvironment and the intra- and inter-tumoral heterogeneity. In the present review, we address the molecular characterization of SCCA and discuss potential diagnostic, predictive and prognostic biomarkers of this complex and challenging disease.

## 1. Introduction

Neoplasms of the anal canal are rare diseases. The most common histology is squamous cell carcinoma (SCCA), responsible for 80% of the cases. Less frequent subtypes include adenocarcinoma, neuroendocrine tumors, malignant melanoma, lymphomas, and mesenchymal tumors [1,2]. In the United States (US), 9090 new cases of SCCA were expected in 2021 [3,4] and the annual incidence of SCCA continues to increase by 2.7% yearly, with a mortality rate rising by 3.1% [5,6,7].

The main risk factors for SCCA development are immunosuppression and Human Papillomavirus (HPV) infection, present in 95% of the cases [8]. Additional risk factors include an increased number of sexual partners and anoreceptive intercourse, which may be linked to HPV infection [9].

The HPV viral DNA integration into the host cells is crucial for the progression of preneoplastic lesions to invasive carcinoma. The production of the leading three oncoproteins with stimulatory properties (E5, E6, and E7) causes the inactivation of known tumor suppressors pRb and p53 (Figure 1). In addition, HPV also activates the PI3K/Akt/mTOR pathway, promoting cell proliferation [10,11].

The probability of developing SCCA is increased 19-fold in an HIV-positive patient compared to the general population in the US [10], with a higher chance in those with long-standing disease (≥15 years) versus those infected more recently (≤5 years) [12].

Despite the success of chemoradiotherapy (CRT) in localized SCCA, 5-year disease-free survival (DFS) is only 31% in T4N1-3M0 patients [8,13]. In this scenario, biomarkers have been studied to predict patient outcomes and determine the best therapeutic strategy for each patient. In the metastatic setting, the therapeutic options are limited due to the rarity of this disease and the paucity of randomized trials. Therefore, the development of biomarkers is essential to predict the response to immunotherapy and other targeted therapies in metastatic patients.

The purpose of this article is to review diagnostic, predictive and prognostic biomarkers in SCCA.

## 2. Standard of Care of SCCA

Approximately 85% of patients with SCCA have non-metastatic disease at presentation. Fluoropyrimidine-based CRT has been the cornerstone of the treatment of localized disease for the past four decades [14]. The combination of two cycles of infusional 5-fluorouracil (5FU) plus mitomycin concurrent with 50–54 Gy of radiation therapy (RT) yields approximately 90% of complete response rate at 6 months [15,16]; however, CRT is associated with meaningful toxicity, mainly cutaneous, gastrointestinal, and hematological adverse events [16,17]. Cisplatin has demonstrated similar efficacy compared with mitomycin, with a lower rate of hematological toxicity, but with limited use in elderly and frail patients [16,17]. Capecitabine has also been equivalent to infusional 5FU and may be associated with both mitomycin and cisplatin. Advances in radiation therapy (RT) techniques, mainly intensity-modulated RT (IMRT), have allowed substantial improvement in cutaneous and gastrointestinal toxicities.

T3-4 and/or node-positive disease are risk factors for disease recurrence, which portends substantial morbidity and mortality; it is estimated that approximately 20% to 44% of SCCA patients will present disease recurrence in a long-term follow-up [13,16,17,18]. Local recurrence may be managed with abdominoperineal resection, which has a meaningful, negative impact on the quality of life, and systemic recurrence typically presents a dismal prognosis, with median OS estimated in 12 months to 20 months [19].

Systemic therapy of metastatic or recurrent SCCA is an unmet clinical need. The first randomized clinical trial addressing the first-line therapy of advanced SCCA recently demonstrated the superior OS and the more favorable toxicity profile of carboplatin plus paclitaxel compared to 5FU plus cisplatin [19]. Modified DCF (docetaxel, cisplatin and 5FU) may be considered for selected patients based on high efficacy but with increased toxicities [20]. There is no standard therapy for chemorefractory patients. Recent phase II studies demonstrate promising results of immune checkpoint inhibition alone or in combination with other therapeutic strategies, with ORR ranging from 12% to 24% [21,22,23,24,25,26]; these studies have not suggested a predictive role of PD-L1 expression for the benefit of immunotherapy in SCCA. A recently presented randomized clinical trial did not show the benefit of the addition of atezolizumab to carboplatin plus paclitaxel in first-line therapy of advanced disease [27]. Other ongoing clinical trials addressing other immunotherapeutic approaches are eagerly awaited in the next few years.

## 3. Molecular Characterization of SCCA

Given the relatively rare incidence of SCCA, comprehensive characterization of its molecular landscape has been scarcely reported when compared to other malignancies [28,29,30]. In the largest study using a multiplatform approach in 199 samples of SCCA, multidrug resistance-associated protein 1 (MRP1) and (epidermal growth factor receptor) EGFR were found to be the most frequent proteins expressed by immunohistochemistry (IHC) at 97.6% and 88.5% respectively [28]. The expression of MRP1 may lead to chemotherapy resistance via drug efflux [31], while the role of EGFR expression remains to be determined. Anecdotal cases of response to anti-EGFR agents have been reported [32], but more data certainly is needed; however, Smaglo and colleagues did not encounter HER2 or PD-L1 expression by IHC, which could be potential biomarkers [28].

In terms of mutational profile, the previously mentioned study revealed 32.6% of PIK3CA mutations in SCCA by next-generation sequencing (NGS) [28]. Similarly, the other two comprehensive genomic profiling studies of SCCA, although with smaller sample sizes, also demonstrated high rates of PIK3CA mutations [29,30], ranging from 29.1 to 40%. Indeed, PIK3CA mutation is the most frequent genetic abnormality in SCCA, being in line with other HPV-related tumors, such as head and neck and cervical cancer. PIK3CA amplification has also been commonly described in SCCA [29,30]. Other frequently mutated genes encountered in SCCA were MLL2 (16–22%) and MLL3 (39%), both of which are important to histone methylation [29,30]. In addition, other genes which are important to DNA damage repair (p53, ATM, HUWE1, BRCA 1, BRCA 2), chromatin remodeling (EP300, SMARCB1, SMARCA4), and activation of Wnt/βcatenin signaling (FAM123B) were also reported as frequently mutated [28,29,30].

Interestingly, mutations in oncogenes critical to the MAPK pathway were not routinely detected in SCCA [28,29,30]. The rare frequency of those mutations, such as KRAS, NRAS, BRAF, EGFR, and HER2, are also rarely found in other HPV-related malignancies, as well as the usual low tumor mutation burden (TMB), with a mean number of 2.5–3.5 somatic mutations/Mb. In addition, HPV-negative SCCA has a different genomic profile as compared to HPV-positive tumors. For instance, loss-of-function mutations in TP53 and CDKN2A are significantly enhanced in patients with HPV-negative disease; these findings are expected since HPV-driven oncogenesis functions through mechanisms of p53 inactivation and cell cycle deregulation via inactivation of Rb [33].

## 4. Biomarkers

### 4.1. HPV

Due to the high incidence of HPV infection in SCCA, HPV detection could presumably be applied as a biomarker for both early diagnosis and treatment response in locally advanced and metastatic disease.

Recent systematic review evaluated the performance of anal cytology, anal HPV testing, p16 or p16/Ki-67 dual staining, and HPV E6/E7 messenger RNA (mRNA) testing for the detection of both anal premalignant and malignant lesions; this study reported a pooled sensitivity and specificity analysis for detecting high-grade anal intraepithelial neoplasia (considered to be a precursor of invasive SCCA) or more advanced lesions of 91.3% (95% CI 78.9–96.7%) and 33.1% (95% CI 22.2–46.3%), respectively [34]; moreover, genotyping for HPV16/HPV18 can potentially improve the specificity of HPV testing for anal premalignant detection.

The authors also showed that HPV E6/E7 mRNA testing for the detection of premalignant lesions and anal cancer, including both HIV-positive and HIV-negative patients, demonstrated a pooled sensitivity and specificity of 74.3% (95% CI 68.3–79.6%) and 65.5% (95% CI 58.5–71.9%), respectively [34]; however, to substantiate the use of screening tests in clinical practice, more studies are needed in both HIV-positive and HIV-negative patients. In addition, there are currently no US Food and Drug Administration-approved tests for anal cancer screening or surveillance.

HPV16 is the most frequently encountered genotype in SCCA, detected in approximately 68% to 94% of cases [35,36,37,38]. The evaluation of tumor HPV16 viral load (HPV16 VL) has demonstrated prognostic value in SCCA patients treated with CRT. An analysis of the HPV16 VL in patients treated with CRT and who presented with a low initial HPV16 VL (detected by PCR) showed significantly less local control (univariate *p* = 0.023, multivariate *p* = 0.042) and overall survival (OS) (univariate *p* = 0.02, multivariate *p* = 0.03) when compared with patients with a higher initial HPV16 VL [39]; moreover, the combined analysis of HPV16 VL and p16INK4a protein expression revealed a significant correlation with decreased local failure.

In terms of prognosis, using HPV as a biomarker is still controversial in clinical practice. HPV+ tumors and p16 positivity have been associated with better local control at five years among SCCA patients [40]. Recent systematic review and meta-analysis of 17 retrospective cohort studies demonstrated that HPV+ and p16+ tumors had improved OS compared to tumors with negative profiles [41]. On the other hand, a prospective cohort with multivariable analyses of non-metastatic patients treated with standard CRT did not show a difference in 5-year OS according to HPV status [42]. Currently, there are no recommendations for treatment modification based on HPV or p16 status (Table 1).

### 4.2. Circulating Tumor DNA (ctDNA)

ctDNA is a small proportion (< 1%) of an individual’s total circulating cell-free DNA and represents a potential non-invasive tool to assess tumor dynamics and characteristics, including tumor response to local and systemic therapies [43]. HPV DNA appears to assess the presence of ctDNA in SCCA and can be detected in plasma with sensitivity up to 93% in HPV-positive cancers [44]. A proof-of-concept study investigated the sensitivity and the prognostic impact of HPV ctDNA detection in 33 histologically proven HPV16-positive or HPV18-positive SCCA treated with CRT. Among 29 patients with detectable HPV ctDNA, its level was associated with lymph node status, with higher median ctDNA levels in node-positive vs. node-negative disease. In 18 post-treatment patients, ctDNA evaluation was available, and residual HPV ctDNA after CRT was associated with shorter DFS (*p* < 0.0001). The three patients with residual HPV ctDNA levels experienced metastatic disease [45]. The Epitopes-HPV02 trial demonstrated that ctDNA was detectable in 91% (52 of 57) of baseline samples in metastatic patients. Residual HPV ctDNA at chemotherapy completion was associated with shorter post-chemotherapy progression-free survival (PFS) and poor one-year OS (11).

### 4.3. HIV

Data regarding clinical outcomes of HIV+ SCCA patients are scarce. Studies on the prognostic and predictive impact of HIV status report conflicting results [42].

An evaluation of linked data from HIV and cancer registries in nine US areas (1996 to 2012) reported that anal cancer incidence was much higher in HIV+ patients compared to the general population (standardized incidence ratio, 19.1; 95% CI, 18.1 to 20.0). In the HIV+ population, anal cancer incidence was highest among men who have sex with men (MSM), increased with age, and was almost four times higher in patients with Acquired Immunodeficiency Syndrome (AIDS) than in those without AIDS (adjusted incidence rate ratio, 3.82; 95% CI, 3.27 to 4.46) [46]. Among these individuals, a low CD4 count was associated with an increased risk of anal cancer, and immunosuppression is critical in the early stages of SCCA development due to the persistent infections and decreased clearance of anal HPV [46,47,48,49,50]; moreover, in a large French cohort study of HIV+ patients, anal cancer incidence increased with the duration of time individuals spent with a low CD4 count and poorly controlled HIV infection [47].

A prolonged cumulative period of immunodeficiency or high viral replication might allow the accumulation of genetic changes that are crucial for developing anal cancer in HIV+ patients. Large randomized controlled trials [51,52,53] have shown the clinical benefit of early antiretroviral therapy (ART) initiation, including the reduction of infection-related cancers [54], and guidelines indicate that ART should be administered to all people living with HIV irrespective of their CD4 counts [55]. In addition, a 2015 meta-analysis of observational studies evaluating the incidence of malignancies before and after the introduction of highly active antiretroviral therapy (HAART) demonstrated that the risk of anal cancer was four times higher in the post-HAART period than in the pre-HAART period [56].

More recently, a systematic review and meta-analysis of 122 studies with data on 417,006 people living with HIV on the association of HIV-related exposures (ART, HIV-RNA plasma viral load (PVL), and CD4 cell count) with outcomes of anal high-risk HPV prevalence, incidence, and persistence. Prevalence, incidence, progression, or regression of anal histological and cytological abnormalities and anal cancer incidence were studied [57]. The results indicated that people living with HIV who receive ART have a decreased prevalence of high-risk HPV, and those with undetectable HIV viral load have decreased risk of high-grade anal lesions (HSIL-AIN2+) prevalence. Overall, ART was not associated with anal cancer risk, but the subgroup of ART users with sustained undetectable HIV viral load had a 44% reduced risk of anal cancer; moreover, a high nadir CD4 cell count of ≥200 cells per µL was associated with a 67% decreased risk of anal cancer incidence [57], however, it is important to point out that the positive effect of ART may not be accurately measured due to the heterogeneity in the history of immunodeficiency in ART users, especially for individuals who might have started HIV therapy before combination ART was introduced (a boosted protease inhibitor or non-nucleoside analog with two reverse transcriptase inhibitors) or according to older guidelines when combination ART was started at low CD4 counts.

Therefore, more recently, people living with HIV are unlikely to experience prolonged periods of immunosuppression, or none at all, which, along with sustained undetectable HIV PVL, could lead to lower rates of anal cancer incidence [57]; however, it is essential to highlight that access to HAART and anal cancer screening/monitoring is far from universally equal and uncertain. Therefore, future prospective studies are needed to monitor anal cancer incidence in people living with HIV globally in the era of universal and early ART.

It is important to note that randomized trials of SCCA have consistently excluded HIV+ patients [16]. In the pre-ART era, CRT was typically applied with reduced doses in early-stage SCCA HIV+ patients due to concerns for increased toxicities [58], and when standard doses of CRT were used, higher rates of grades 3/4 adverse events were reported; those treatment issues correlated with inferior outcomes in HIV+ patients [59].

With the advent of ART and its consequent immune restoration, more HIV+ patients have been treated with standard CRT doses. Nonetheless, while some studies show that HIV+ patients have comparable disease control and OS to HIV-negative patients [60,61,62,63,64,65], others suggest increased treatment-related toxicity and/or decreased local control [66,67,68,69]. In a prospective cohort, HIV positivity was associated with a worse response to CRT (OR, 5.72; 95%CI, 2.5–13.0; *p* < 0.001). In multivariate analyses, the absence of complete response at 6 months was the main factor associated with inferior OS (HR 3.36, *p* = 0.007, 95% IC 1.39–8.09). The 5-year OS rate was 62.5% in the HIV+ group compared to 78.0% among HIV− patients with a median follow-up of 66 months, although this difference was not statistically significant (*p* = 0.40) [22].

In the metastatic setting evaluating systemic chemotherapy, only recent studies have included HIV+ patients. The phase II InterAAct trial, which randomized SCCA patients to either first-line therapy with cisplatin and 5-fluorouracil or carboplatin and paclitaxel, included HIV+ patients provided they were on HAART, and CD4 count was ≥200/μL or with undetectable plasma HIV-positive VL [19].

Immune checkpoint inhibitors (ICI) have been extensively used in several tumor types; however, in SCCA, ICI have been evaluated with mild efficacy, with limited inclusion of HIV+ patients [70]. Therefore, efforts to evaluate immunotherapy strategies in HIV+ patients with advanced solid tumors are ongoing (NCT02408861 and NCT02595866). Immune response against tumors, HPV, and HIV, is a complex interaction within T cells, cytokines, and immune checkpoints. The knowledge is still limited about the intricacy of immune system operation during HPV, HIV, and tumor subsistence [71,72].

### 4.4. Ki-67

Ki-67 protein is a nuclear antigen found in proliferating cells and has been described as a potential biomarker for predicting outcomes in anal cancer [73].

Ki-67 was reported as an independent predictor of DFS, demonstrating that a higher Ki-67 index correlated with improved DFS in patients treated with CRT [74]; moreover, the evaluation of EGFR and Ki-67 protein co-expression ratio in pretreatment tumor biopsies of patients treated with CRT indicated a positive correlation between EGFR expression and Ki-67 on immunohistochemistry. Furthermore, the study demonstrated that patients presenting a higher ratio (Ki-67 expression and EGFR staining) had a statistically significant lower OS, disease-free failure, and a higher risk of locoregional failure when compared to patients with a lower ratio, suggesting a possible role of this ratio as a possible biomarker and also raising questions of benefits regarding anti-EGFR therapy [75]. Of note, in the same study, considering tumor Ki67 and EGFR as single markers, no statistically significant differences were seen in any clinical endpoints.

More recently, in a prospective cohort of patients treated with CRT, Ki-67 was not a useful marker to predict response, PFS or OS [42]. Therefore, it is logical to presume that a cellular proliferation marker such as Ki-67 could be a useful predictive or prognostic biomarker in anal cancer; however, studies have reported conflicting results, and the practical and clinical application remains uncertain.

### 4.5. PD-L1

As previously mentioned, the majority of patients with SCCA present detectable levels of HPV. Of note, HPV viral proteins E6 and E7 contribute to the oncogenic transformation of SCCA into invasive cancer and can also induce an antitumor host response by recruitment of tumor-infiltrating lymphocytes. Some retrospective studies have evaluated the programmed cell death protein ligand 1 (PD-L1) expression in tumor cells of patients with SCCA. Overall, PD-L1 positivity varied from 56% to 68.8% [76,77,78,79,80]. The prognostic role of the biomarker has been reported in those studies, with conflicting results [76,77,78,79].

Immune checkpoint blockade with monoclonal antibodies against programmed cell death protein-1 (PD-1) and PD-L1 has also been investigated in metastatic SCCA with encouraging results; however, not all studies demonstrated the application of PD-1/PD-L1 values in tumor specimens as a marker of response. A multicenter phase II trial NCI 9673 (Part A) studied 37 patients with treatment-refractory metastatic SCCA treated with nivolumab monotherapy [70]. The overall response rate (ORR) was 24%, with two complete responses and seven partial responses. The median PFS was 4.1 months, and the median OS was 11.5 months; however, in this study, patients were not preselected or stratified based on PD-1/PD-L1 positivity.

Single-agent pembrolizumab was also studied in advanced SCCA treatment and KEYNOTE-028, a multicohort phase Ib trial for patients with PD-L1 positive advanced solid tumors [23], enrolled 24 patients with PD-L1-positive SCCA patients, presenting an ORR of 17%, with four partial responses and a median PFS of 3.0 months and a median OS of 9.8 months. In 2020, Marabelle and colleagues conducted a multicohort phase II KEYNOTE-158 study enrolling 112 patients with SCCA [24]. The presented ORR was 11.6%, with the median duration of response was not reached. Responses occurred in 14.7% of patients with PD-L1 combined positive scores ≥ 1 and 6.7% with PD-L1 combined positive scores < 1. Among all patients, the median PFS was 2.0 months, and the median OS was 12.0 months. Therefore, these findings suggest a predictive role of PD-L1 combined score for tumor response to immunotherapy.

A recent presentation on a novel checkpoint inhibitor, retifanlimab, an anti-PD-1 monoclonal antibody, demonstrated an ORR of 13.8%, a PFS of 2.3 months, and a median OS of 10.1 months for treatment-refractory SCCA regardless of PD-L1 expression [25]. Although the positive results demonstrated in the aforementioned trials were not uniform regarding PD-L1 status as a predictor of response, both single-agent pembrolizumab and nivolumab have been incorporated in NCCN and ESMO guidelines for refractory metastatic SCCA regardless of PD-L1 expression, but they have not yet been approved by regulatory agencies [4,81].

Regarding the non-metastatic setting, a small Chinese cohort study [82], 5 female patients (4 of them with PD-L1 positive tumors) with locally advanced SCCA were treated with four cycles of neoadjuvant PD-1 antibody toripalimab combined with docetaxel and cisplatin, followed by radiotherapy and two cycles of concurrent toripalimab. All of them achieved a cCR after neoadjuvant treatment. Currently, there are at least 4 undergoing clinical trials investigating the role of immunotherapy in locoregional SCCA (NCT04719988, NCT05374252, NCT05060471 and NCT03233711).

Therefore, PD-1/PD-L1 status has not proven to be a reliable biomarker to predict patient response to treatment with immune checkpoint inhibitors, and, as such, the question remains on how to better select patients for immunotherapy.

### 4.6. Tumor-Infiltrating Lymphocytes (TILs)

It is well known that host immune recognition can interfere with an anti-tumor response. Regarding HPV-positive tumors, such as the vast majority of SCCA, host immune response against viral neoantigens, characterized by the presence of cytotoxic CD8+ tumor-infiltrating lymphocytes (TILs), may prompt response to both CRT and immunotherapy and disease control [83,84]. In other HPV-related tumors, such as HPV-associated oropharyngeal cancer, patients with HPV-positive/TIL-high had a 3-year OS of 96% vs. 59% in the HPV-positive/TIL-low group [85].

A retrospective analysis of 284 patients treated with radiation therapy with or without chemotherapy for locally advanced non-metastatic SCCA analyzed the prognostic value of TILs in this tumor subtype [86] and demonstrated that, in p16+ patients, tumors with absent/low levels of TILs presented a relapse-free rate of 63%, compared to 92% of patients with high levels of TILs (*p* = 0.006). Yet another study, prospective in nature, evaluated the prognostic value of CD8+ TILs in 150 patients treated with CRT and reported that higher values of CD8+ TILs predicted an improved local control (*p* = 0.023) [79]. The authors also describe findings of patients with high PD-1+ TILs expression presented better local control (*p* = 0.003), DFS (*p* = 0.007) and OS (*p* = 0.039) [79]; this data reinforces the knowledge that HPV-positive tumors with an exuberant immune cell infiltration present a more favorable outcome, including treatment with immunotherapy.

### 4.7. RAS/BRAF

RAS and BRAF proteins are components of the epidermal growth factor receptor (EGFR) downstream signaling pathway and their gain-of-function mutations lead to constitutive activation of the RAS/RAF/MAPK pathway, contributing to cell survival and proliferation [87].

Mutations in the exons 2, 3, and 4 of the RAS isoforms (mainly KRAS, and less frequently NRAS, and HRAS) and in the codon 600 of the BRAF protein (V600E) are well-known prognostic and predictive biomarkers in colorectal cancer (CRC) [88]. Metastatic CRC patients who harbor these mutations do not derive benefit from the use of anti-EGFR monoclonal antibodies cetuximab and panitumumab [87]. More recently, it was demonstrated that previously treated BRAF V600E-mutated CRC patients may benefit from the combination of the BRAF inhibitor encorafenib plus the anti-EGFR cetuximab [89].

Comprehensive molecular characterization studies have demonstrated that both RAS and BRAF mutations are infrequent in patients with SCCA [28,29,30]. There are no prospective studies evaluating the role of anti-EGFR monoclonal antibodies in both localized and advanced disease. Retrospective studies suggest the efficacy of these agents in advanced SCCA, both in monotherapy and in combination with chemotherapy [32,90,91,92]. The randomized phase II CARACAS trial explored dual PD-1 and EGFR blockade in previously treated advanced SCCA patients. Avelumab alone was compared with the combination of avelumab plus cetuximab in 60 patients. The ORR, the primary endpoint, was 10% versus 17%, respectively. Median PFS was 2.05 months versus 3.88 months in a median follow-up of 11.0 months, and grade 3–4 adverse events were 13.3% versus 33.3%, respectively [26].

In the absence of prospective studies demonstrating the benefit of anti-EGFR therapy or BRAF inhibitors in SCCA, RAS and BRAF mutations should not be routinely tested in the management of these patients.

### 4.8. Tumor Mutational Burden (TMB)

SCCA has been characterized as having a low tumor mutational burden (TMB), with median TMB reported at 2.5 mutations/megabase (Mb) on whole-exome sequencing [93]. Other authors have reported the prevalence of TMB-high in SCCA ranging from 8% (TMB ≥ 17 mutations/Mb) to 13.6% (TMB ≥ 10 mutations/Mb), with the overwhelming majority of cases harboring low TMB [94,95]. Upon analyzing ORR for patients with TMB-high SCCA treated with pembrolizumab in the KEYNOTE-158 trial (a trial that led to FDA approval for pembrolizumab in solid tumors with TMB ≥ 10 mutations/Mb), it was reported no improvement in ORR (7% vs. 11%) for patients with TMB-high versus non–TMB-high SCCA, respectively [96]. Therefore, the utility of TMB as a predictive biomarker for the benefit of anti–PD-1 monotherapy has not proven helpful in this context, and, as in most gastrointestinal tumors, TMB cannot be applied as a reliable biomarker for treatment guidance.

### 4.9. Microsatellite Instability

The presence of microsatellite instability-high (MSI-H) among patients with SCCA is infrequent. Of the 223 patients with non-colorectal MSI-H tumors included in the KEYNOTE-158 trial, which evaluated pembrolizumab in this population, there was only one SCCA patient [24]. Despite the rarity of MSI-H SCCA, the agnostic role of this molecular target in predicting response to immunotherapy in several tumor types compels oncologists to search for this biomarker regardless of its extremely low prevalence.

## 5. Conclusions

SCCA is a rare malignant neoplasm etiologically linked to HPV infection, with common molecular characteristics compared to the other HPV-related malignancies, such as cervical and head and neck cancers; however, comprehensive molecular characterization studies indicate that SCCA presents specific epigenomic, genomic, and transcriptomic abnormalities which suggest that targeted therapies and genome-guided personalized therapies should be specifically designed for this disease. Actionable mutations are rare in SCCA and immune checkpoint inhibition has not yet been proven useful in an unselected population of patients. Therefore, advances in systemic therapy of SCCA will only be possible with the identification of predictive biomarkers and the development of targeted therapies or immunotherapeutic approaches that consider the unique tumor microenvironment and the intra- and intertumoral heterogeneity of this complex and challenging disease.

## Figures and Tables

**Figure 1 biomedicines-10-02029-f001:**
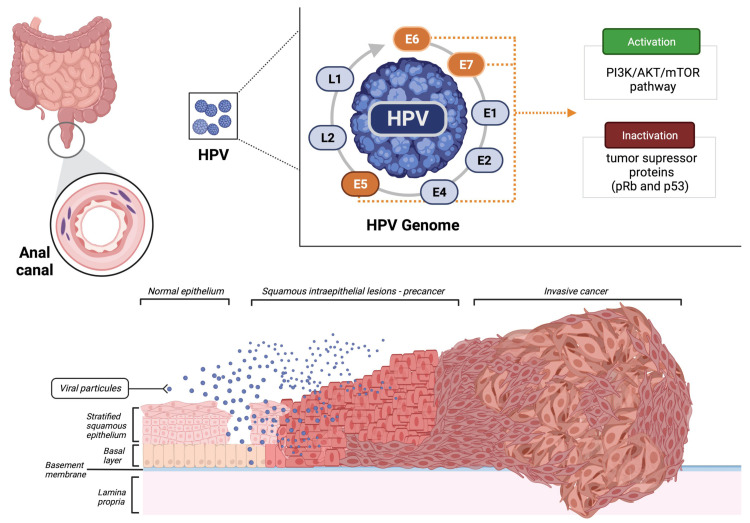
Malignant transformation of the normal epithelium of the anal canal by the human papillomavirus (HPV). Created using BioRender.

**Table 1 biomedicines-10-02029-t001:** Key Points.

Squamous cell carcinoma of the anal canal is a rare neoplasm, but with rising incidence and is extremely related to prior HPV infection.
The probability of developing anal cancer increases with HIV infection especially in long-standing disease.
Despite the success of local therapies, some patients present with disease recurrence and survival in the advanced disease setting is disappointing.
HPV can be applied as a biomarker for both early diagnosis and treatment response in locally advanced and metastatic disease.
Residual HPV ctDNA after CRT treatment can be associated with worse outcomes.
The role of PD-L1 expression and TMB is yet to be defined.
Microsatellite instability although extremely rare can predict response to immunotherapy.

## Data Availability

Not applicable.

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
