# Peer review of "Biomarkers in Anal Cancer: Current Status in Diagnosis, Disease Progression and Therapeutic Strategies"

_biomedicines, 2022, doi:10.3390/biomedicines10082029_

Round 1
Reviewer 1 Report
Dear Authors,
Thank you for submitting your interesting research.
Comments:
The manuscript has been well organized and discussed.
Adding some key points before the conclusion may make the study results more attractive.
Author Response
“The manuscript has been well organized and discussed. Adding some key points before the conclusion may make the study results more attractive.”
Thank you so much for the suggestion, we have added a box of Key Points to our paper to the manuscript as per attached document.
Reviewer 2 Report
The manuscript reviews biomarkers in anal cancer. The authors rightfully state that locally advanced anal cancers have a dismal prognosis and there is certainly an unmet need for improvements of treatment. Biomarkers can play an important role in order to stratify patients. The manuscript is clearly written and gives a good overview over the current state of treatment results. Adding HIV to biomarkers like HPV and ctDNA seems a bit strange but i can see the merits of this and the paragraph is an interesting read.
The omission of tumor infiltrating lymphocytes as a separate subsection of the biomarkers needs to be adressed. There are many research paper who focused on TIL (using HE staining or CD8 IHC). With regards to PD-L1 i would suggest to briefly discuss the few works who analyzed PD-L1 in pretreatment samples.
I would also suggest to discuss the current landscape of trials that assess PD-L1 inhibition in non-metastatic anal cancer (there is a british, american and german trial as far as i am aware).
Author Response
Reviewer #2:
“The manuscript reviews biomarkers in anal cancer. The authors rightfully state that locally advanced anal cancers have a dismal prognosis and there is certainly an unmet need for improvements of treatment. Biomarkers can play an important role in order to stratify patients. The manuscript is clearly written and gives a good overview over the current state of treatment results.”
- “Adding HIV to biomarkers like HPV and ctDNA seems a bit strange but i can see the merits of this and the paragraph is an interesting read.”
Thank you so much for the response. Regarding HIV as a biomarker, we found that it is also an interesting point of review considering the prevalence of HIV positivity in this population.
- “The omission of tumor infiltrating lymphocytes as a separate subsection of the biomarkers needs to be adressed. There are many research paper who focused on TIL (using HE staining or CD8 IHC). With regards to PD-L1 i would suggest to briefly discuss the few works who analyzed PD-L1 in pretreatment samples.”
Thank you for the extremely pertinent suggestion. We have included a section on TILs in our manuscript. However, research papers regarding TILs in colorectal cancer, but less data regarding TILs in anal cancer is available.
- “I would also suggest to discuss the current landscape of trials that assess PD-L1 inhibition in non-metastatic anal cancer (there is a british, american and german trial as far as i am aware).”
Thank you for the suggestion, we have included information regarding previous and ongoing trials of PD-L1 inhibition in the non-metastatic scenario.